# Meibomian Gland Dysfunction: Intense Pulsed Light Therapy in Combination with Low-Level Light Therapy as Rescue Treatment

**DOI:** 10.3390/medicina57060619

**Published:** 2021-06-14

**Authors:** Leonidas Solomos, Walid Bouthour, Ariane Malclès, Gabriele Thumann, Horace Massa

**Affiliations:** Clinic of Ophthalmology, Geneva University Hospitals & Faculty of Medicine, University of Geneva, CH-1205 Geneva, Switzerland; walid.bouthour@hcuge.ch (W.B.); Ariane.malcles@hcuge.ch (A.M.); gabriele.thumann@hcuge.ch (G.T.); horace.massa@hcuge.ch (H.M.)

**Keywords:** dry eye disease, meibomian gland dysfunction, intense pulsed light, low-level light therapy

## Abstract

*Background and Objectives:* Evaporative dry eye disease is frequently associated with meibomian gland dysfunction. Patients are often unhappy because of daily drops, care burden, and suboptimal conventional treatments. In this study, we assessed the efficacy of a novel device, the Eye-light^®^, a combination of intense pulsed light therapy and low-level light therapy, as a novel treatment for meibomian gland dysfunction and dry eye disease. *Materials and Methods:* This was a retrospective, single-center study carried out over a 6-week period, in which 22 eyes from 11 patients were included. Each patient received four combined light therapy treatment sessions, once weekly over 4 weeks. Patients underwent a clinical examination and filled out a standardized questionnaire to evaluate symptoms one week prior to treatment, and one week after the fourth session. *Results:* Combined light therapy improved several ocular surface outcome measures in our patients. This study demonstrates that this adjunctive treatment significantly improves the ocular surface and quality of life of patients with dry eye disease and meibomian gland dysfunction. *Conclusions:* Combined light therapy may be included in meibomian gland dysfunction treatment protocols as an adjunctive rescue treatment.

## 1. Introduction

Meibomian gland dysfunction (MGD) is a major risk factor for dry eye disease (DED) through tear evaporation [1]. DED and MGD induce inflammation of both the ocular surface and the eyelid margin and severely alter the quality of life of a large part of the population. DED and MGD lead to frequent ophthalmological consultations, making them a substantial economic burden [2]. Symptoms include a more or less intense foreign body sensation, and burning in the eyes. Conventional treatments for MGD and subsequent DED include warm lid compresses, lid scrubs, topical antibiotic or antibiotic/steroid drops, and systemic antibiotic treatment [3].

Recently, novel technologies have emerged to improve meibomian gland function in MGD and alleviate related symptoms of DED. Intense pulsed light (IPL) therapy is one of these. Initially FDA-approved as a treatment of telangiectasia in acnea-rosacea, IPL has been increasingly studied in dry eyes, as it has proven to be effective in relieving ocular symptoms. As a matter of fact, telangiectasia are also a hallmark of MGD. In addition to the effect of IPL on telangiectasia, which could account for its long-term effect (up to 6 months) [4], there is mounting evidence that IPL relieves the obstruction of the meibomian gland ductules and orifices and reduces inflammation [5,6]. The intense pulses of non-coherent light delivered in a wide range of wavelengths are thought to act on various chromophores present in the skin such as hemoglobin or melanin, through a local heating effect.

Another novel treatment is low-level light therapy (LLLT). LLLT induces tissue photobiomodulation through light-emitting diodes (LEDs). LEDs offer the advantage of acting above the skin (in the optical tissue window), as the sources emit in the range of infrared to visible wavelengths (500 to 1100 nm). Photoactivation with a specific wavelength induces cell activation and changes in inflammatory protein expression in the periorbital tissue [7,8]. As ocular surface and eyelid inflammation is a hallmark of MGD, LLLT photobiomodulation may be an interesting therapy in MGD.

The Eye-light^®^ device combines IPL and LLLT and has recently obtained CE marking for combined light therapy. The procedure consists of two steps: first, IPL is applied to the inferior eyelid and lower periorbital region. Then, LLLT is applied over the entire periorbital area covering both upper and lower eyelids, which remain shut throughout the procedure [9]. Combined light therapy may result in both the softening of meibum, thus relieving obstruction of the glands, and the stimulation of the glands themselves. Since it has been reported in a recent Cochrane analysis that IPL therapy for MGD shows only scarce evidence of efficacy [10], we hypothesized that combined IPL and LLLP therapy may result in both softening of meibum, which relieves obstruction of the glands, and stimulation of the glands.

The aim of this study was to retrospectively assess the clinical efficacy of combined light therapy on meibomian gland function on the ocular surface quality, and the quality of life of patients suffering from MGD and DED.

## 2. Materials and Methods

This was a retrospective, single-center study carried out over a 6-week period, at the University Hospitals of Geneva, Switzerland. Every patient provided a written informed consent allowing the use of their anonymized clinical data for research purposes.

### 2.1. Procedures

We performed combined light therapy (IPL therapy followed by LLLT) using the Eye-light^®^ device. Treatment was carried out following the instructions in the manufacturer’s user manual.

Patients sat in an armchair. The treatment consisted of IPL followed by LLLT. IPL treatment consisted of five painless light shots. The first shot was applied just below the lower eyelid nasally, and two other shots were applied contiguously by moving the probe temporally. The fourth spot was applied at the lateral canthus. Finally, the last spot was applied in a horizontal manner along the inferior orbital rim. The same sequence was applied to the contralateral eye. The IPL treatment for both eyes lasted no more than 5 min. Each shot was applied on a spot of 2.5 × 4.5 mm with a wavelength of 600 nm. Patients wore protection goggles provided by the manufacturer throughout the IPL treatment. Subsequently, the LLLT mask was applied for 15 min, with a wavelength of 633 nm, immediately after the IPL procedure. During LLLT, relaxing music was played to increase patient comfort. No protective goggles were required during LLLT, and patients were asked to keep their eyes shut, in order to allow for a complete treatment of both upper and lower eyelids.

### 2.2. Study Design

Each patient received four combined light therapy treatment sessions, once weekly over 4 weeks. We treated both eyes simultaneously during each session. Patients underwent a clinical examination and filled out a standardized questionnaire to evaluate symptoms (Ocular Surface Disease Index, OSDI) one week prior to treatment, and one week after the fourth session.

We documented treatments before and during the protocol. There was no standardized treatment regimen prior to the study. Any prior treatment for MGD remained unchanged throughout the duration of the study. The main inclusion criterion for the study was the association of MGD and failed therapies. Patients were considered for analysis if they had symptoms of DED in association with abnormal lid margin findings and a poor meibum score [11]. Patients were not considered for treatment if they were pregnant or had a medical condition which could preclude the use of IPL and LLLT such as a previous malignant tumor of the skin. Patients were excluded from final analysis (i) if their baseline treatment was altered; (ii) if they did not complete all four treatment sessions; or (iii) if they were lost to follow-up.

### 2.3. Outcomes

We used the MGD grading scale [12] for clinical evaluation. The MGD scale grades the following parameters: abnormal lid margin findings of vascularity, plugging of gland orifices, lid margin irregularity, lid margin thickening, partial glands, and gland dropout. The meiboscore was assessed using pictures of the upper and lower eyelids taken by the infrared module of the Spectralis HRA+OCT device (Heidelberg Engineering inc., Heidelberg, Germany). Three trained physicians evaluated the gland deficit and were blinded to each other’s results. The median score was obtained based on the previous publication from Reiko Arita et al. [12]. We also evaluated the tear break-up time (TBUT), corneal fluorescein staining (Oxford scheme), and Schirmer test with an anesthetic (Schirmer 1 test).

### 2.4. Statistical Analysis

We verified data distribution normality with the Shapiro–Wilk test. We applied pairwise Student’s t-test whenever data were distributed normally. Whenever data were not distributed normally, we used the non-parametric Wilcoxon matched pairs signed-rank test and reported the sum of signed ranks W, and the *p*-value. Significance threshold was 0.05. We used GraphPad Prism (v9.1.0) [13] for statistics and figures.

## 3. Results

Included in the study were 22 eyes from 11 patients (6 women and 5 men, age 52.6 ± 6.5 years). Two patients were lost to the post-treatment follow-up.

The ongoing and past treatments of MGD are listed in Table 1.

OSDI score. We found a significant difference in the OSDI score before (mean = 33.73, SEM = 5.79) and after treatment (mean = 19.15, SEM = 3.47); *t* (16) = 2.16, *p* < 0.05 (Figure 1A).

MGD score. Before treatment, the median MGD score was 12 points, and 10.5 points after treatment. The difference was significant, W = −99, *p* < 0.005 (Figure 1B).

Clinical tests. We found a significant increase in the tear break-up time before (median = 4.5 s) and after treatment (median = 6.5 s), W = 131, *p* < 0.005 (Figure 1C). The difference was also significant in the corneal fluorescein staining score before (median = 1) and after (median = 1), W = −21, *p* < 0.05 (Figure 1D). There was no significant difference before and after treatment (i) in the corneal fluorescein staining score (median before = 1, median after = 1; Mann–Whitney U = 114.5, *p* = 0.123) (Figure 1D); (ii) and in the Schirmer 1 test (median before = 12, median after = 12; W = 3, *p* = 0.96) (Figure 1E).

Meiboscore. There was a significant difference in the meiboscore before (median = 2) and after treatment (median = 1.5), W= −21, *p* < 0.05 (Figure 1F).

In a subgroup analysis, we found significant differences before and after treatment in the lid margin vascularity (median before = 3, median after = 2, W = −36, *p* < 0.01) (Figure 2A), plugging of gland orifices (median before = 3, median after = 2; W = −45, *p* < 0.01) (Figure 2B), and partial glands (median before = 2, median after = 2, W = −21, *p* < 0.05) (Figure 2E). We observed no significant difference in irregularity, thickening, and gland dropout after treatment (Figure 2C,D,F).

## 4. Discussion

The data presented here demonstrate that treatment with IPL combined with LLLT significantly improved the OSDI score, TBUT, corneal fluorescein staining score, the meiboscore, and the MGD score and alleviated meibomian gland dysfunction. To our knowledge, there is only one other study that assessed the efficacy of a combination of IPL and LLLT [9].

Our data confirm the results of Stonecipher and colleagues [9] showing that combined light treatment with the Eye-light^®^ device improves the ocular surface quality in patients with MGD. At one week after the 4 weekly treatments, we observed a 44% decrease in the OSDI score, whereas Stonecipher and colleagues found a decrease of up to 57% after one treatment; however, it was not specified how long after treatment the OSDI score was determined. Stonecipher and colleagues reported an increase in TBUT from 4.4 to 8.0 s after treatment, which is larger than the 4.5 to 6.5 s increase we observed. Even though we found a smaller increase in TBUT, 6.5 s is in the low range of normal [14]. All clinical outcomes directly related to MGD were improved by combined light therapy. As expected, the Schirmer 1 test remained unchanged.

Previous studies using monthly IPL therapy without LLLT have shown that four treatment sessions were required to obtain a statistically significant increase in TBUT [15]. Our data show that four treatments, administered once weekly, improve the symptoms of MGD considerably; however, we only followed the patients for one week after the end of the protocol and therefore do not know how long the benefits last. In our subgroup analysis of the MGD score, we found no difference in the lid margin irregularity and thickening and in gland dropout before and after treatment, possibly because these are chronic signs that might only change in the long run. Therefore, further studies on the effect of combined light treatment over several months on these parameters are warranted. Albietz et al. treated patients with IPL every two weeks and noted improvement two weeks after the third treatment. This may suggest that weekly treatments might offer a quicker clinical effect [16].

In our study, we did not observe any side effect. Previous studies reported discomfort, redness, or swelling in up to 13% of treated patients [15].

We should note that the patients included in our study continued topical treatment and eyelid hygiene throughout the duration of the study; however, since all patients were suffering from dry eye disease as determined by the OSDI score, the treatments were not effective but may have influenced the results. Since treatment with IPL combined with LLLT improved ocular surface symptoms, notwithstanding the effect of pre-procedure treatment, IPL combined with LLLT may be an effective adjunct to current pharmacological therapies. However, the efficacy of IPL alone or as an add-on treatment remains unclear [17].

Recent animal studies showed promising results in rabbits [18]. One potential explanation is the decrease in pro-inflammatory cytokines such as TNF alpha and IL-6 [19]. One possible mechanism of light therapy may be the neutralization of TNF alpha and thus a reduction in ocular surface inflammation in dry eye patients [20]. Interestingly, this effect is dose-dependent and might explain the rapid improvement of the ocular surface condition in our patients [21].

Another potential mechanism of action is heat tissue penetration combined with a constant LLLT exposition for 15 min. The light wavelength (633 nm) emitted by the mask is sufficient to penetrate at a depth of one to two millimeters into the eyelid tissue [22]. This suggests a possible mechanism of action of LLLT on the meibomian glands.

Our study has several limitations. Firstly, we did not compare our results to a control or sham group because of the retrospective design of our protocol. A prospective, randomized, and controlled trial should be performed to assess the efficacy of combined light therapy alone or as an add-on treatment. The assessment of longer periods of treatment, and chronic effects of combined light therapy should provide blueprints to optimize the treatment protocol. Intrapatient variability due to subjective ratings might have skewed our results. This is especially problematic when dealing with dry eye disease. Thus, the interpretation of the OSDI score should be carried out carefully. Nevertheless, this is counterbalanced by the fact that other objective outcomes yielded significant improvement.

## 5. Conclusions

In conclusion, IPL combined with LLLT as an adjunct treatment improves the ocular surface and quality of life of patients with DED and MGD, with treatment once weekly over 4 weeks. Thus, combined light therapy may be included in MGD treatment protocols as an adjunctive rescue treatment for those patients.

## Figures and Tables

**Figure 1 medicina-57-00619-f001:**
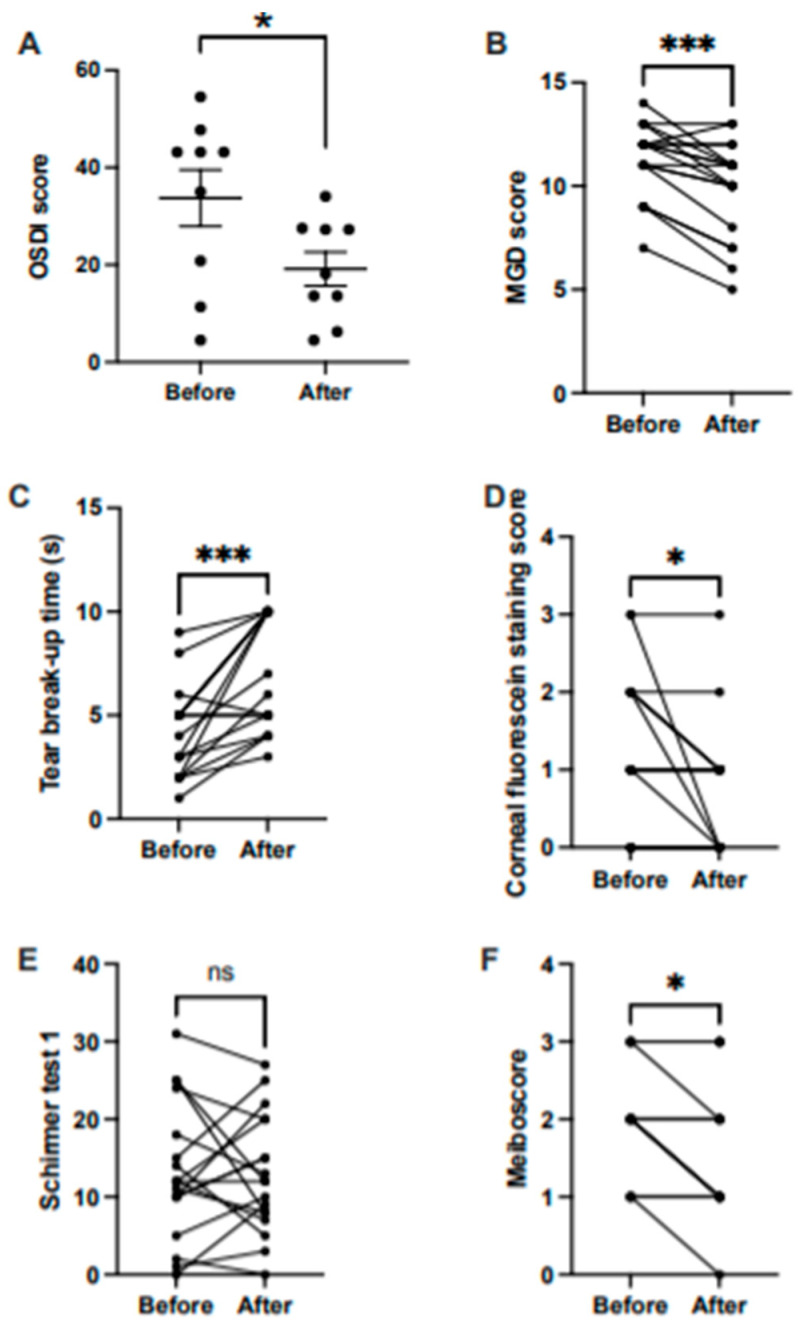
Treatment effect before and after treatment on (**A**) OSDI score (horizontal bars represent means, and error bars represent the standard error of the mean, SEM); (**B**) MGD score; (**C**) tear break-up time; (**D**) corneal fluorescein staining score; (**E**) Schirmer 1 test; and (**F**) meiboscore. Lines shown in B, C, D, E, and F represent outcomes before and after treatment. s, seconds. * *p* < 0.05; *** *p* < 0.005; ns, non-significant.

**Figure 2 medicina-57-00619-f002:**
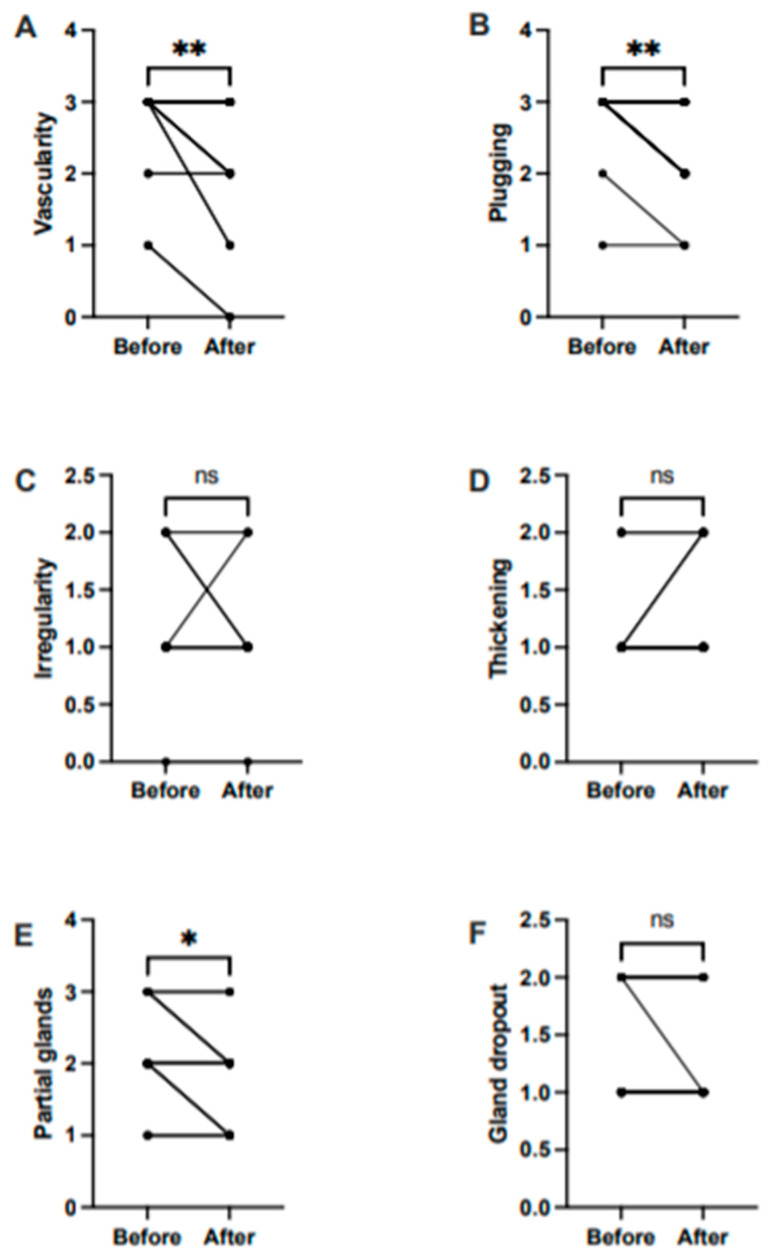
Treatment effect before and after treatments on components of the MGD score. (**A**) Abnormal lid margin findings of vascularity; (**B**) plugging of gland orifices; (**C**) lid margin irregularity; (**D**) lid margin thickening; (**E**) partial glands; and (**F**) gland dropout. Lines represent outcomes before and after treatment. * *p* < 0.05; ** *p* < 0.01; ns, non-significant.

**Table 1 medicina-57-00619-t001:** Summary of ongoing and past treatments for MGD. * in the month before IPL + LLLT treatment.

Treatment	Number of Patients	Percentage
Eyelid hygiene with warm compresses	9/9	100%
Artificial tears	9/9	100%
Ongoing steroids	1/9	11%
Past steroids *	5/9	55%
Ongoing ciclosporine	2/9	22%
Past ciclosporine *	1/9	11%
Past fusidic acid *	3/9	33%
Physiological serum	2/9	22%
Autologous serum	1/9	11%

## Data Availability

All data supporting reported results can be found at the Clinic of Ophthalmology, Geneva University Hospitals.

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
