# Peer review of "Meibomian Gland Dysfunction: Intense Pulsed Light Therapy in Combination with Low-Level Light Therapy as Rescue Treatment"

_medicina, 2021, doi:10.3390/medicina57060619_

Round 1

Reviewer 1 Report

This is a very interesting and novel technique presented by the authors.

1) It would be appropriate to elaborate on the treatment the patients were already on. Was it just artificial tears and lid hygiene? Was it topical corticosteroids, azithromycin or cyclosporin that would contribute to the results? Were they practising warm compresses that would also contribute to the results? Although you do mention that they continued their treatment, a clarification of the treatment would provide more information as per the efficacy of IPL.

2) Line 131-132: " Nevertheless, a TBUT of over 6 seconds is not considered a dry eye disease anymore". Please provide a reference.

3) It would be of value to add a few more information about patient's dry eye, ie the use of systemic medications and more importantly the inclusion criteria that makes them specifically evaporative and not mixed type DED

Reviewer 2 Report

the manuscript presents an interesting study but there are numerous points to improve in order to make it publishable.
the idea of ​​double action is clearly valid but the introduction must be expanded to explain the goal of the manuscript. The application of the device must be better explained in the materials and methods. furthermore, the statistical study is unclear (expert review would be better).
in the discussion / conclusion chapter the conclusion must be more captivating. In addition, the English language needs an important improvement

Round 2

Reviewer 2 Report

the authors have completed the required changes in a very complete way. in progress they have greatly improved the structure of the manuscript.
my opinion is that the manuscript can be published.